# Network Data Maps on Entrepreneurial Intention, Unicorns, and Human Flourishing on the SCOPUS Database: A Visual Analysis Using VOSviewer

**José Manuel Saiz-Alvarez** [1,2,3]

1   Graduate School, Catholic University of Santiago de Guayaquil, Guayaquil 090615, Ecuador
2   Master in Management and Administration in Social Economy and Cooperativism, Catholic University of Avila, 05005 Ávila, Spain
3   Study and Research Center "Enzo Faletto", University of Santiago de Chile, Santiago 8350575, Chile

**Abstract:** Using the SCOPUS database and VOSviewer, this paper aims to analyze the bibliographic information on three keywords (entrepreneurial intention (EI), human flourishing (HF), and unicorns) to identify relevant areas for current and future research on entrepreneurship by applying a bibliometric and content review approach to 2434 documents for the BMA (business, management, and accounting) and EEF (economics, econometrics, and finance) subject areas to construct and visualize bibliometric networks on the basis of co-citation and co-authorship relations in these items. The main findings of this study are as follows: (1) the number of documents published in the European Union on EI (600) almost doubles those published on this topic in the United States (354); the United States leads the number of papers (113) published on HF, and the number of documents published on Unicorns by BRICS (Brazil, Russia, India, China, and South Africa) countries (22) almost equals the number of documents published on this issue in the United States (25); (2) research on EI during the core years of the COVID-19 pandemic (2019–2022) is of growing interest linked to entrepreneurship education and psychological traits; (3) ethics-related entrepreneurial behavior has historically supported current HF-related research; (4) entrepreneurial ecosystems, leadership, and innovation are critical success factors for born globals to be unicorns; (5) there is a geographic disparity (Spain, India, and the US) in the most cited authors for EI, HF, and unicorns, respectively.

**Keywords:** entrepreneurial intention; human flourishing; unicorn; ethics; COVID-19; SCOPUS; born global



## 1. Introduction

The COVID-19 pandemic is causing deep economic and business disruptions worldwide, particularly in developing countries. Demand-side disruption happens when leading companies underestimate innovation-based startups, and supply-side disruption occurs when traditional firms do not create new competencies to satisfy clients' demands [1]. Disruptions in supply and demand are linked to financial restrictions and business shortages, leading to inflationary pressures that threaten multiple dimensions of the wellbeing of people. From this point of view, the lagging effects of the pandemic create a high potential to lead to a global recession with a negative impact on organizational sustainability and socioeconomic welfare due to restricted or negative economic growth in a worldwide recession.

Socioeconomic imbalances commonly threaten and can harm business structures in the productive sector and commercial systems of micro, small, and medium enterprises (MSMEs). The effects can have profound implications for business firms in emerging and developing countries where affected business structures can profoundly impact vast numbers of MSMEs and entrepreneurs with a low financial capacity to secure the sustainability of their firms.

Entrepreneurship plays a vital role during these disruptive times caused by COVID-19 to counterbalance unpredictable actions in firms' organizational management by considering a global VUCA (volatile, uncertain, complex, and ambiguous) environment. HF has a pivotal role in many firms in developing countries due to the financial instability caused by national and international business interruption, economic shortages due to confinement, and constant workplace transformation needs.

The pandemic has temporarily affected entrepreneurial ecosystems worldwide. Furthermore, entrepreneurial ecosystems are interdependent actors and coordinated factors to impulse entrepreneurship within a given territory [2]. As a result, entrepreneurial ecosystems are productive structures to encompass complex interactions based on knowledge and innovation, driving economic agents' competitive capabilities [3] to foster business growth and startup creation guided by innovation. Firms use open innovation by intentionally combining internal and external knowledge flows to accelerate internal innovation and market expansion to apply innovations externally [2]. However, a joint analysis of EI, HF, and unicorns is a gap in the literature related to entrepreneurship; hence, constructing network data maps linking EI, HF, and unicorns is justified.

However, one of the main problems for entrepreneurs in developing countries is the difficulty of undertaking successfully, especially by the younger population, given the generally small amounts of capital made available for startup creation with a minimum guarantee of success. After the first year of operation, more than 85% of startups have to close, generally due to a lack of financing and, in the most severe cases, the financial ruin of the entrepreneur. To avoid this severe socioeconomic problem and for entrepreneurship to flourish rapidly, the public administration must intervene to help finance these business projects before they disappear definitively. In general, there is no venture capital in developing countries, and, if present, its existence and effects on the economy and employment are usually almost symbolic.

This public–private collaboration is vital in entrepreneurship education [4] and coopetition [5], as they are crucial factors in building entrepreneurial ecosystems, especially in less developed regions where the presence of HEIs (higher-education institutions) in a municipality helps to attract more startups [6].

Entrepreneurial ecosystems contribute to determining entrepreneurial intention (EI), defined as the entrepreneur's preference, intrinsic cognition, and behavioral tendency to create a startup [7]. This psychological attitude determines entrepreneurial behavior, which is the process via which entrepreneurs put their entrepreneurial vision into practice [8]. In this respect, the authors of [9] showed that intentions are the single best predictor of planned behavior, both conceptually and empirically. They depend on attitudes toward the target behavior, which, in turn, reflect beliefs and perceptions.

When the EI of creating a firm is fulfilled with success, human flourishing (HF) emerges in the individual. Given the growing difficulties in entrepreneurship, especially in economic crises, HF should constantly encourage the entrepreneur over time. Constant encouragement and impulse, on some occasions, are among the psychological keys to creating unicorns.

Given this entrepreneurial context rooted in entrepreneurial ecosystems, this paper aims to implement a bibliometric analysis of EI, HF, and unicorns as determinants, among other factors, for business sustainability, economic growth, and wealth creation in disruptive times. To cope with this goal, VOSviewer version 1.6.8, developed by Nees Jan van Eck and Ludo Waltman from the Centre for Science and Technology Studies (CWTS), Leiden University, the Netherlands, is used with data from the SCOPUS database and some analysis tools from the database. This bibliometric revision is structured as follows: first, it provides an overview of the literature related to the three items to be analyzed (EI, HF, and unicorns), and one proposition for each item is defined; second, it presents the materials and methodology; third, the results and discussion are shown on the basis of the bibliometric analysis; fourth, some conclusions and future research are provided.

## 2. Literature Review

The MSMEs located in developing countries are characterized, in general, by being companies conceived not to create wealth and distribute it in society but for the merely personal and family survival of the entrepreneurs who work in it. Insufficient financial resources, narrow markets, lack of planning, and a long-term vision prevent the company from growing and flourishing. Professional experience is not enough for business success, as it must be complemented with specialized training. The entrepreneur should at least dispose of minimal technical knowledge for the company to survive in hostile environments marked by solid business competition. Only when happiness flourishes at work will human capital benefit the organization by improving performance, productivity, creativity, and organizational citizenship behavior [10,11]. As a result, the organization will be able to compete.

Given these premises, there is a growing interest worldwide, as shown in Table 1, to understand the keys to explaining the entrepreneurial process occurring on the planet.

**Table 1.** Top countries for papers published on EI, HF, and unicorns, limited to the BMA and EEA subject areas (SCOPUS database between 2014 and 2022).

| Entrepreneurial Intention | | | | | |
|---|---|---|---|---|---|
| **Country** | **Documents** | **Country** | **Documents** | **Country** | **Documents** |
| United States | 354 | Malaysia | 158 | South Africa | 95 |
| Spain | 234 | India | 127 | China | 86 |
| United Kingdom | 187 | France | 122 | Indonesia | 83 |
| Germany | 163 | Australia | 104 | Italy | 81 |
| **Human Flourishing** | | | | | |
| **Country** | **Documents** | **Country** | **Documents** | **Country** | **Documents** |
| United States | 113 | Italy | 16 | Germany | 12 |
| United Kingdom | 61 | Spain | 16 | New Zealand | 8 |
| Australia | 23 | Canada | 15 | South Africa | 7 |
| Netherlands | 17 | France | 13 | Finland | 6 |
| **Unicorns** | | | | | |
| **Country** | **Documents** | **Country** | **Documents** | **Country** | **Documents** |
| United States | 25 | Italy | 7 | Portugal | 3 |
| United Kingdom | 13 | Australia | 6 | Brazil | 3 |
| India | 9 | Canada | 4 | Germany | 3 |
| China | 7 | Netherlands | 4 | Russia | 3 |

Legend: BMA (business, management, and accounting), EEA (economics, econometrics, and finance).

The prevalence of specific values affects levels of entrepreneurship and EI [12], as EI is significantly associated with gender, education, entrepreneurial parents, and a proactive personality [13]. Furthermore, Zhao, Hills, and Seibert [14] showed that the effects of perceived learning from entrepreneurship-related courses, risk propensity on EI, and previous entrepreneurial experience are fully mediated by entrepreneurial self-efficacy.

Krueger, Reilly, and Carsrud [15] compared two intention-based models in terms of their ability to predict entrepreneurial intentions: Ajzen's theory of planned behavior (TPB) and Shapero's model of the entrepreneurial event (SEE). According to these authors, Ajzen argued that EI depends on feasibility, perceptions of personal attractiveness, and social norms, contrary to Shapero, who affirmed that EI depends on feasibility, perceptions of personal desirability, and propensity to act.

Recent research has focused on the influence of social networks and the internet on EI. As shown by Pérez-Fernández et al. [16], social network size and the need for achievement

positively influence the entrepreneurial information obtained in social networks, which in turn, positively impacts EI.

Given the crucial importance of EI, proposition 1 (P1) can be set as follows:

**Proposition 1 (P1).** *During the core years of the COVID-19 pandemic (2019–2022), research on entrepreneurial intention linked to entrepreneurship education and psychological traits has been a growing interest.*

Human flourishing (HF) plays a crucial role in sustainability and business survival, considering that knowledge, experience, and skills are needed to transform challenges into opportunities aiming to attain systematic improvement in the wellbeing of people as a condition for wealth creation in business enterprises.

HF relates to education, resilience, and fruitful human resources management (HRM). Related to the relationship between HF and education, Sylveira et al. [17] proposed an inverse relationship between HF and EI, as HF can be enhanced in upper secondary education to boost EI later in students' lives. Academic success in studies tends to be reproduced later in business, as active, energetic, engaged, and focused employees provide a sustainable competitive advantage to the firm [18]. As a result, first-order competitive advantages can benefit organizational leadership.

Added to HF, Wakil, Sun, and Chan [19] proposed a "co-flourishing" framework integrating community resilience and tourism development by mobilizing six types of community capital—human, social, natural, physical, financial, and psychological—which strengthen community capacity during disturbances or crises. Societies endowed with these six types of community capital are more prone to be successful and enduring, mainly when organizations compete in "glocalized" business environments.

Globalization has fostered necessity and opportunity entrepreneurship worldwide, where HF is the final result of organizational investments and commercial and productive activities. As a result, resilience and sustainability play crucial roles in this challenging process. In this sense, Alcaraz et al. [20] outlined an externally oriented model (centered on corporate priorities, communities' flourishing, and ecosystems' resilience) to advance HRM and sustainability.

As a result, proposition 2 (P2) can be defined as follows:

**Proposition 2 (P2).** *The entrepreneurial behavior related to ethics mainly supports the current development of HF.*

Unicorn startups are closely related to HF in entrepreneurship science. The concept of a unicorn refers to organizations that suddenly flourish propelled by teams of highly specialized people who start a privately owned company with a valuation of at least one billion USD before launching IPO (initial public offerings) in the stock markets.

As of 2018, there were 261 unicorns worldwide, of which 68 were from China, mainly in Beijing (40), Shanghai (15), and Shenzhen (7) [21]. Only 3 years later, the global number of unicorns was 1024, of which 487 were established in the United States and 301 in China (Table 2), with the tech company ByteDance being the highest-valued Unicorn worldwide, with a total value of 350 billion USD (Table 3).

**Table 2.** Number of global unicorns in 2021, by country.

| | | In % | | | In % | | | In % |
|---|---|---|---|---|---|---|---|---|
| United States | 487 | 47.6 | Canada | 15 | 1.5 | Mexico | 5 | 0.5 |
| China | 301 | 29.4 | Brazil | 12 | 1.2 | Switzerland | 4 | 0.4 |
| India | 54 | 5.3 | South Korea | 10 | 1.0 | Sweden | 4 | 0.4 |
| United Kingdom | 39 | 3.8 | Indonesia | 7 | 0.7 | Spain | 3 | 0.3 |
| Germany | 26 | 2.5 | Singapore | 7 | 0.7 | Netherlands | 3 | 0.3 |
| France | 19 | 1.9 | Japan | 6 | 0.6 | | | |
| Israel | 17 | 1.7 | Australia | 5 | 0.5 | | | |

Source: adapted from [22].

**Table 3.** Leading top 10 unicorns worldwide as of 2021 (in billion USD).

| Company | Country | Valuation | Company | Country | Valuation |
|---|---|---|---|---|---|
| ByteDance | China | 350 | Canva | Australia | 40 |
| ANT Group | China | 150 | Instacart | United States | 39 |
| SpaceX | United States | 100 | Databricks | United States | 38 |
| Stripe | United States | 95 | Cainiao | China | 34 |
| Klarna | Sweden | 46 | Revolut | United Kingdom | 33 |

Source: adapted from [22].

The number of unicorns in China and other regions outside the US has risen recently, whereas the phenomenon was initially limited to the US [23]. This strong growth of unicorns is based on the application of technology in born global companies. In this regard, digital technologies, such as artificial intelligence, big data, and the Internet of things, are becoming increasingly mature, profoundly impacting Industry 4.0, and representing the driving force behind a new wave of innovation and entrepreneurship-related activities worldwide [24].

Intellectual capital, composed of the sum of human capital, relational capital, and structural capital [25], is crucial in creating and boosting unicorns. In this regard, the principals of unicorns hire human capital capable of taking higher than normal risks with their investment to disrupt a given market and succeed [26].

**Proposition 3 (P3).** *Entrepreneurial ecosystems, leadership, and disruptive innovation are critical success factors for born globals to be unicorns.*

Unlike national startups, born globals benefit from brand new relationships with a group of heterogeneous international partners [27]. These relationships open commercial flows among organizations to satisfy stakeholders, especially clients. The increasing degree of digitization, accelerated by the pandemic, leads to market globalization, regardless of the company's geographical location. These new business opportunities benefit organizations in developing countries by competing with significantly lower prices than those of the competition in developed countries with higher production costs of the products and services offered to the market.

**3. Methodology**

To test these propositions, the author combines the SCOPUS data analysis tool with VOSviewer, version 1.6.18, a software tool for constructing and visualizing bibliometric networks built on the basis of citation, co-occurrence, bibliographic coupling, co-citation, or co-authorship relations. VOSviewer was chosen given its characteristics to develop and view two-dimensional distance-based maps, regardless of the mapping technique used to construct the map [28] on the basis of co-occurrence and co-authorship. As research protocol, this paper applies a bibliometric and content review approach in investigating

current resilience research in the items EI, HF, and unicorns to identify relevant areas for future research limited to BMA and EEF subject areas only. These terms were chosen because EI in unicorns has a crucial role in achieving HF. The text mining functionality offered in VOSviewer is used to construct and visualize EI, HF, and unicorn co-occurrence networks.

The first requirement in the bibliometric analysis is to delineate the relevant source concepts to identify the items to be included in the review [29,30]. This requirement was complemented by analyzing co-occurrences and total link strengths, and visual analysis was conducted and displayed using network visualization and overlay visualization. The co-occurrence network was used to reveal the hotspots and the research landscape [31] within the chosen items EI, HF, and unicorns as part of entrepreneurship as a research topic. A similar analysis was conducted for co-citation among authors.

## 4. Results

This section shows the findings of the bibliometric review. The subsections overview the three VOSviewer outputs: keyword co-occurrences and total link strengths, network visualization, and overlay visualization for the specific keywords EI, HF, and unicorns.

Seminally published by [9], the item "EI" has attracted the attention of researchers, with up to 3528 document results from 27 subject areas published from 2014 onward, as shown in the SCOPUS database, of which 2092 were in the BMA (business, management, and accounting) and EEF (economics, econometrics, and finance) subject areas (Table 4).

**Table 4.** Documents in the SCOPUS database for 2014–2022 (till August) limited to the BMA and EEF subject areas.

| Keywords | 2022 | 2021 | 2020 | 2019 | 2018 | 2017 | 2016 | 2015 | 2014 | Total |
|---|---|---|---|---|---|---|---|---|---|---|
| EI | 248 | 396 | 305 | 268 | 216 | 227 | 155 | 153 | 124 | 2092 |
| HF | 31 | 35 | 23 | 43 | 25 | 32 | 15 | 21 | 19 | 244 |
| Unicorns | 11 | 20 | 13 | 13 | 14 | 11 | 9 | 7 | 0 | 98 |
| *Total* | 290 | 451 | 341 | 324 | 255 | 270 | 179 | 181 | 145 | 2434 |

Legend: BMA (business, management, and accounting), EEF (economics, econometrics, and finance), EI (entrepreneurial intention), HF (human flourishing).

A total of 2434 documents, from the years 2014 to 2022 (until August), including papers (1973 | 81.06%), book chapters (217 | 8.91%), conference papers (141 | 5.79%), reviews (36 | 1.48%), books (30 | 1.23%), and others (37 | 1.52%), were identified in the SCOPUS database (Table 5), of which 57.71% were published after the emergence and dissemination of COVID-19 (2019–present), which shows a growing interest in researching these issues.

**Table 5.** Type of SCOPUS documents for 2014–2022 (till August) limited to the BMA and EEF subject areas.

| Keywords | Papers | Book Chapters | CP | Reviews | Books | Others | Total |
|---|---|---|---|---|---|---|---|
| EI | 1755 | 149 | 126 | 30 | 10 | 22 | 2092 |
| HF | 150 | 56 | 10 | 5 | 16 | 7 | 244 |
| Unicorns | 68 | 12 | 5 | 1 | 4 | 8 | 98 |
| *Total* | 1973 | 217 | 141 | 36 | 30 | 37 | 2434 |

Legend: BMA (business, management, and accounting), CP (conference papers), EEF (economics, econometrics, and finance), EI (entrepreneurial intention), HF (human flourishing).

### 4.1. EI: Co-Occurrences

Co-occurrence among papers was analyzed (Table 6), settling on 20, the minimum number of occurrences of the 4355 keywords, of which 25 met the threshold. These items

were grouped into five clusters with 170 links and 618 total link strength, as shown in Figures 1 and 2.

**Table 6.** EI: Co-occurrences and total link strength of business-related selected keywords.

|  | OC | TLS |  | OC | TLS |  | OC | TLS |
|---|---|---|---|---|---|---|---|---|
| ENTED | 200 | 655 | HE | 36 | 116 | Creativity | 27 | 79 |
| Attitude | 60 | 250 | Innovation | 29 | 112 | ES | 18 | 74 |
| Motivation | 61 | 207 | Personality | 25 | 94 | NAC | 16 | 59 |

Legend: ENTED (entrepreneurship education), ES (entrepreneurial skills), HE (higher education), NAC (need for achievement), OC (occurrences), TLS (total link strength).

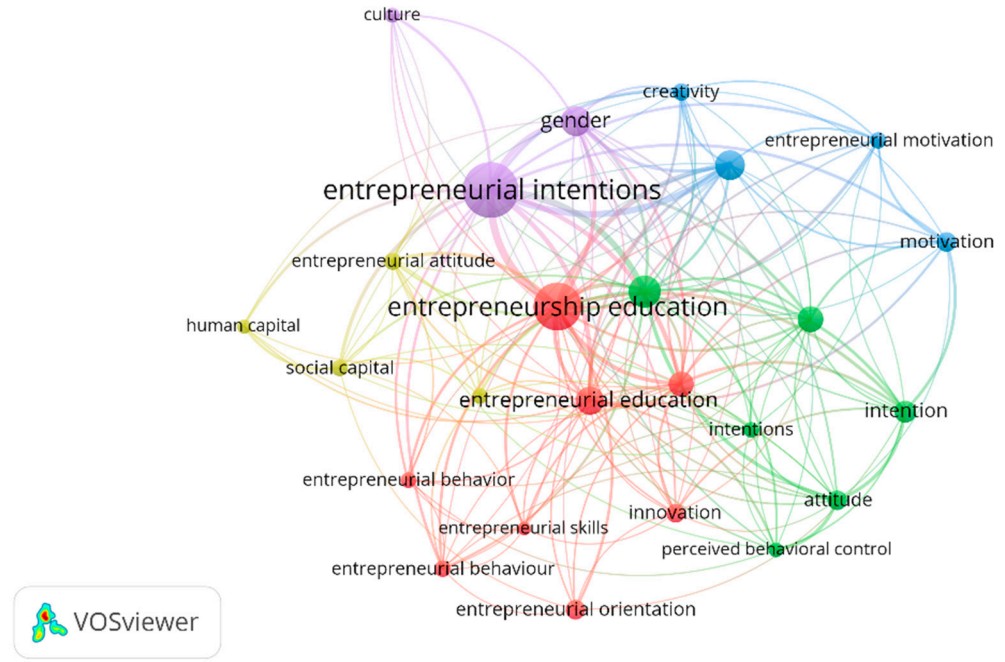

**Figure 1.** Network visualization (entrepreneurial intention—SCOPUS). Cluster 1 (Green), Cluster 2 (Red), Cluster 3 (Blue), Cluster 4 (Yellow), Cluster 5 (Violet).

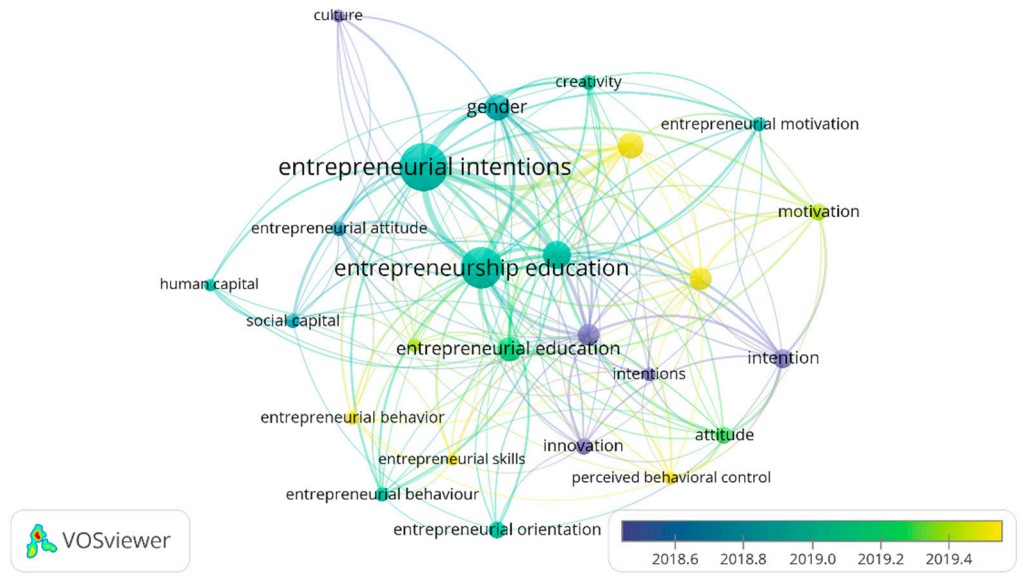

**Figure 2.** Overlay visualization (entrepreneurial intention—SCOPUS).

*4.2. EI: Co-Citations*

Co-citation among documents was analyzed, setting on 300 the minimum number of citations for an author. Of the 73,336 authors, 90 met the threshold. For each of the 90 authors, the total strength of the co-citation links with other authors was calculated, and the top 10 most cited authors and clusters they belong to are shown in Table 7.

**Table 7.** EI: Co-citation and total link strength of the top 10 most cited authors.

|  | CL | CC | TLS |  | CL | CC | TLS |
|---|---|---|---|---|---|---|---|
| Liñán, F. | 2 | 3066 | 115,427 | Kolvereid, I. | 2 | 1207 | 50,506 |
| Ajzen, I. | 2 | 2586 | 89,029 | Bandura, A. | 4 | 1136 | 43,044 |
| Krueger, N.F. | 2 | 1899 | 71,866 | Kautonen, T. | 2 | 889 | 36,211 |
| Fayolle, A. | 3 | 1728 | 66,245 | Van Gelderen, M. | 2 | 787 | 34,130 |
| Carsrud, A.L. | 2 | 1252 | 48,453 | Urbano, D. | 5 | 778 | 30,040 |

Legend: CC (co-citations), CL (cluster), TLS (total link strength).

These 90 items were grouped into five clusters (Table A1) with 4005 links and 984,740 total link strength, as shown in Figure 3, where clusters are grouped in colors.

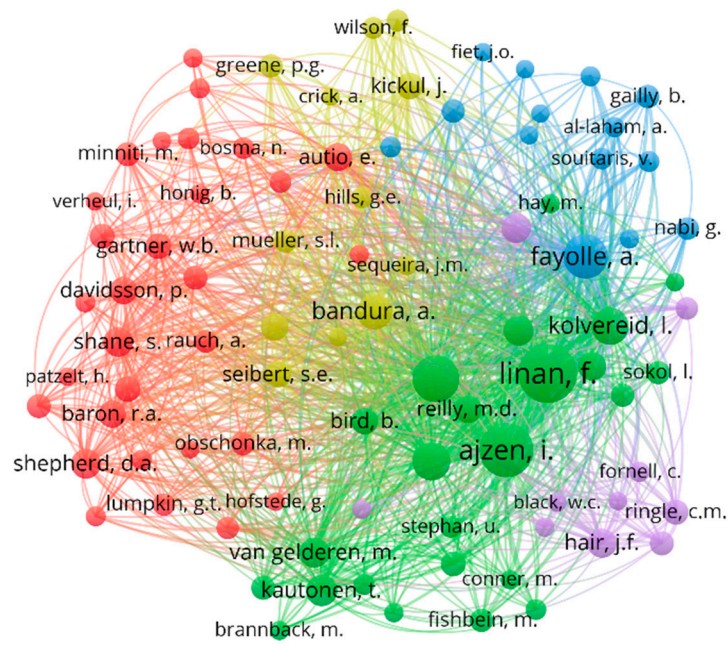

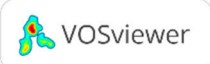

**Figure 3.** Co-citation visualization (entrepreneurial intention—SCOPUS). Cluster 1 (Red), Cluster 2 (Green), Cluster 3 (Blue), Cluster 4 (Yellow), Cluster 5 (Violet).

*4.3. HF: Co-Occurrences*

Regarding the item "human flourishing", there were 4075 document results, of which 244 were in the BMA and EEF subject areas (Tables 4 and 5). A co-occurrence analysis was conducted, with five being the minimum number of occurrences of the 1354 keywords, of which 26 met the threshold, with 16 being the most extensive set of related items, of which nine are shown in Table 8.

**Table 8.** HF: Co-occurrences and total link strength of business-related selected keywords (SCOPUS database).

|  | OC | TLS |  | OC | TLS |  | OC | TLS |
| --- | --- | --- | --- | --- | --- | --- | --- | --- |
| Ethics | 26 | 41 | Virtue ethics | 11 | 21 | Education | 7 | 14 |
| Sustainability | 15 | 11 | Wellbeing | 11 | 17 | Meaningful work | 7 | 9 |
| Leadership | 11 | 16 | PSP | 8 | 13 | HM | 6 | 5 |

Legend: HM (humanistic management), OC (occurrences), PSP (positive psychology), TLS (total link strength).

These 16 keywords were grouped into five clusters (1: green, 2: red, 3: blue, 4: yellow, 5: violet), 37 links, and 54 total link strength (Figures 4 and 5).

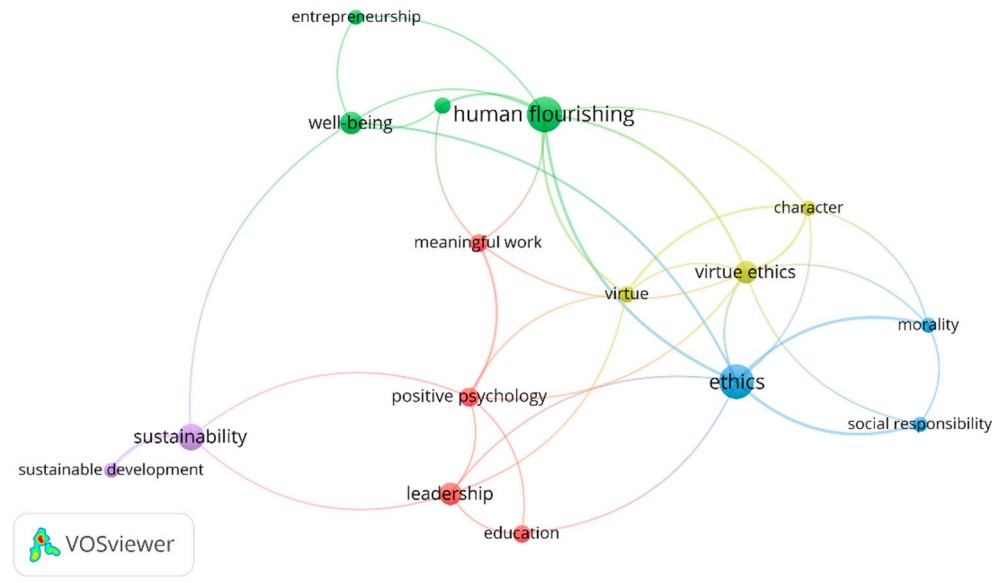

**Figure 4.** Network visualization (human flourishing—SCOPUS).

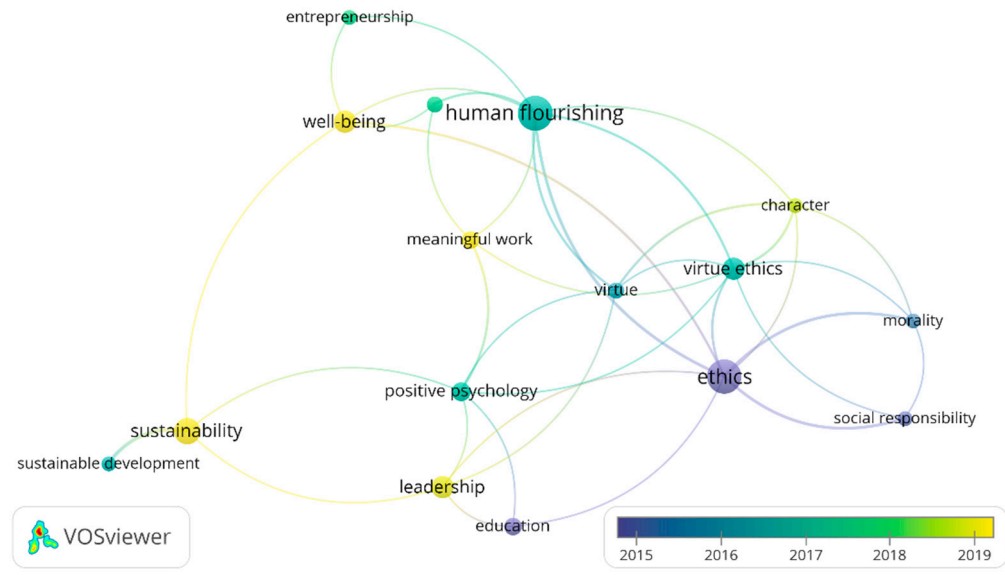

**Figure 5.** Overlay visualization (human flourishing—SCOPUS).

*4.4. HF: Co-Citations*

Regarding authors and citations on HF, co-citation among documents was analyzed, setting on 20 the minimum number of citations for an author. Of the 24,188 authors, 72 met

the threshold. For each of the 72 authors, the total strength of the co-citation links with other authors was calculated for authors with more than 50 citations (Table 9).

**Table 9.** HF: Co-citation and total link strength of the top 10 most cited authors.

|  | CC | TLS |  | CC | TLS |
|---|---|---|---|---|---|
| Sen, A.K. | 225 | 10,377 | Diener, E. | 73 | 3308 |
| Nussbaum, M.C. | 118 | 4277 | Seligman, M.E.P. | 73 | 2010 |
| MacIntyre, A. | 96 | 1413 | Ryan, R.M. | 66 | 2491 |
| Csikszentmihalyi, M. | 91 | 3840 | Dweck, C.S. | 65 | 311 |
| Ruger, J.P. | 89 | 5199 | Deci, E.L. | 58 | 1630 |

Legend: CC (co-citations), TLS (total link strength).

These 72 items were grouped into five clusters (Table A2) with 1321 links and 55,677 total link strength, as shown in Figure 6.

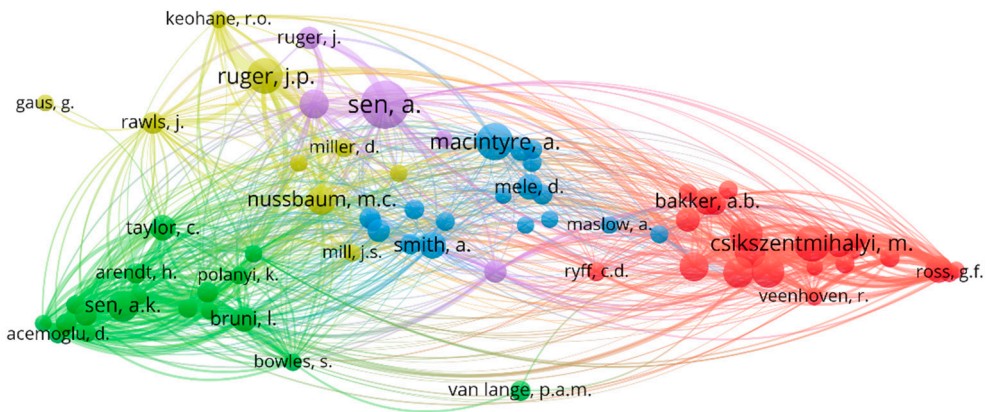

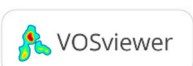

**Figure 6.** Co-citation visualization (human flourishing—SCOPUS). Cluster 1 (Red), Cluster 2 (Green), Cluster 3 (Blue), Cluster 4 (Yellow), Cluster 5 (Violet).

*4.5. Unicorns: Co-Occurrences*

Concerning the item "unicorns", 1180 document results were found in the SCOPUS database, of which 98 were in the BMA and EEF subject areas (Tables 4 and 5). A co-occurrence analysis was conducted, with three being the minimum number of occurrences of the 453 keywords, of which 15 met the threshold, but three were discarded to avoid redundancy (Table 10).

**Table 10.** Unicorns: Co-occurrences and total link strength of business-related selected keywords (SCOPUS database).

|  | OC | TLS |  | OC | TLS |  | OC | TLS |
|---|---|---|---|---|---|---|---|---|
| VC | 13 | 36 | Startups | 5 | 20 | Ecosystem | 3 | 10 |
| ENT | 12 | 33 | DI | 2 | 12 | Commerce | 2 | 8 |
| Innovation | 6 | 21 | ID | 2 | 12 | Leadership | 4 | 7 |

Legend: DI (disruptive innovation), ENT (entrepreneurship), ID (innovation diffusion), OC (occurrences), TLS (total link strength), VC (venture capital).

These 12 keywords were grouped into three clusters (Table 11), with 28 links and 47 total link strength (Figures 7 and 8).

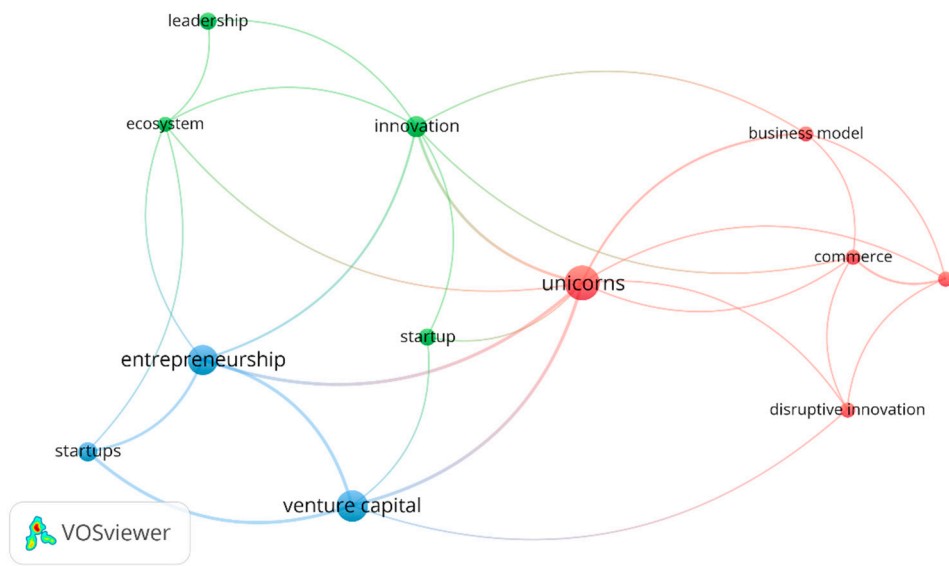

**Figure 7.** Network visualization (unicorns—SCOPUS). Cluster 1 (Green), Cluster 2 (Red), Cluster 3 (Blue).

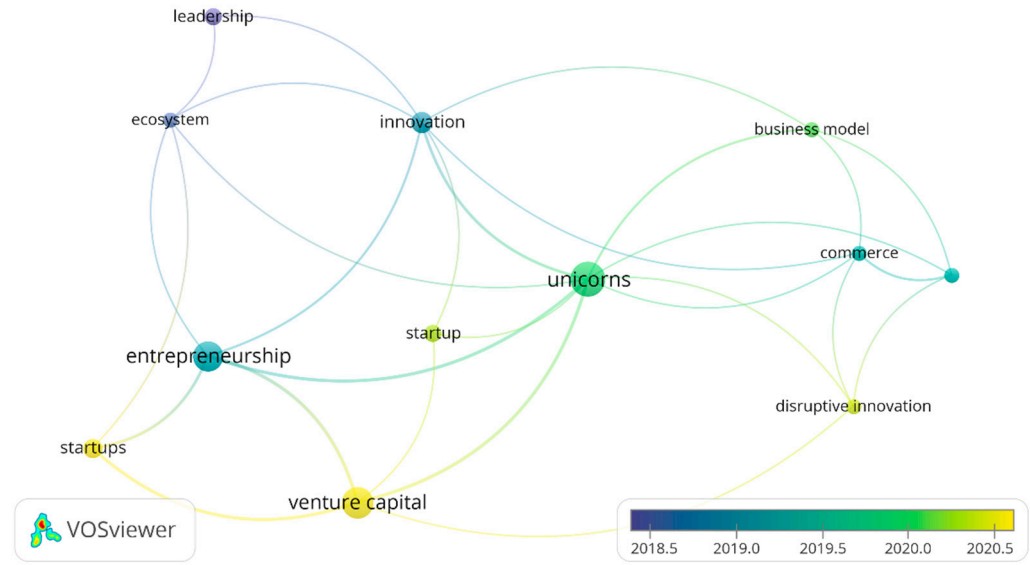

**Figure 8.** Overlay visualization (unicorns—SCOPUS).

### 4.6. Unicorns: Co-Citations

Regarding authors and citations on unicorn-related literature on SCOPUS, co-citation among documents was analyzed, setting on 20 the minimum number of citations for an author. Of the 5936 authors, six met the threshold. For each of the six authors, the total strength of the co-citation links with other authors was calculated (Table 11).

**Table 11.** Unicorns: Co-citation and total link strength of the top 10 most cited authors.

|  | CL | CC | TLS |  | CL | CC | TLS |
|---|---|---|---|---|---|---|---|
| Audretsch, D.B. | 1 | 37 | 185 | Wrigley, C. | 2 | 20 | 40 |
| Kuratko, D.F. | 2 | 21 | 161 | Lerner, J. | 1 | 25 | 28 |
| Acs, Z.J. | 1 | 24 | 100 | Moro Visconti, R. |  | 25 | 0 |

Legend: CC (co-citation), CL (cluster), TLS (total link strength).

These five items were grouped into two clusters (1: Red, 2: Green) with seven links and 257 total link strength, as depicted in Figure 9.

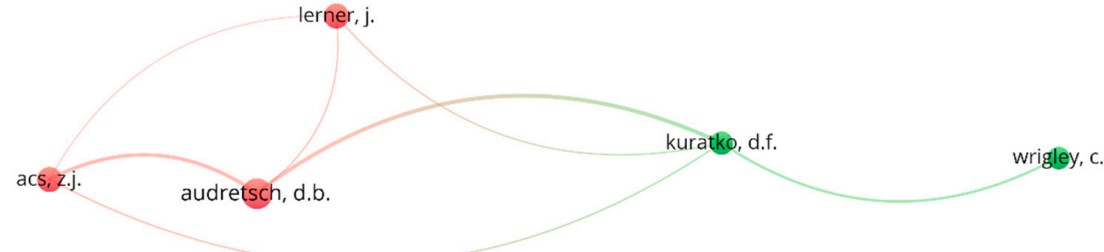

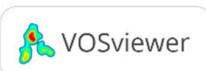

**Figure 9.** Co-citation visualization (unicorns—SCOPUS).

## 5. Discussion

### 5.1. The Importance of Soft Skills in EI

Regarding P1, as only 21 documents were published in the 1970s, 1980s, and 1990s, EI as a research topic became popular from 2015 onward, accounting for 81.6% of all publications in SCOPUS limited to the BMA and EEF subject areas. As a result, 1909 (54.11%) documents were published between 2019 and 2022 (Table 12). This result shows the higher interest by researchers in topics related to entrepreneurship due to the impact generated by the COVID-19 pandemic.

**Table 12.** SCOPUS documents published on EI (2014–2022).

| Period | 3 Year Average | Year | Documents |
| --- | --- | --- | --- |
| 1 | 144 | 2014 | 124 |
| | | 2015 | 153 |
| | | 2016 | 155 |
| 2 | 237 | 2017 | 227 |
| | | 2018 | 216 |
| | | 2019 | 268 |
| 3 | 316.3 | 2020 | 305 |
| | | 2021 | 396 |
| | | 2022 | 248 |
| TOTAL | | | 2092 |

Note: Until August 2022.

Furthermore, 3 year averages showed an accelerating growing interest in researching EI during the years of the COVID-19 pandemic. Compared to 2014–2016, the 3 year average multiplied by 2.19 during the COVID-19 pandemic, as shown in Table 13.

Especially in disruptive times, soft skills "enable individuals to interact harmoniously with others, create trust, improve productivity, and guide employees" [32] (p. 56). When soft skills are used, HF can be reinforced in the workplace. What determines the difference and the competitiveness of organizations is the human capital working in it, primarily relational capital. As entrepreneurs flourish successfully in the labor market, the relational capital increases, and their value augments in the labor market.

When the exogenous disruption is intense, as with the COVID-19 pandemic, temporary imbalances in the firm tend to become permanent. This fact leads the management

team (or single entrepreneurs in MSMEs) to carry out structural transformations in their organizations guided by corporate strategies that often combine in-house innovation, human talent, agility, and the expansion of market niches. Human capital and skills are critical in second-generation companies, where new managers tend to have a more global and digitalized business vision. This vision leads to more corporate risk, thus giving rise to the succession problem in the organization, mainly between the second and third generations, especially if strategic disruptions are radical.

As presented in Figures 1 and 2, EI closely links to entrepreneurial education and, to a lesser extent, entrepreneurial motivation and innovation. As depicted in Figure 2, the items "education", "intention", and "innovation" were previously more specific aspects related to entrepreneurship. When innovations are fostered within the organizations, intrapreneurs are born, pushing the organization toward entering new market niches and broadening its strategic vision and corporate policies. Disposing of new ideas strengthens the intention to continue offering innovative ways to increase efficiency and productivity in the firm.

These primary links were clustered in aspects related to entrepreneurship, especially in cluster 2, composed of culture, gender, entrepreneurial attitude, entrepreneurial behavior, entrepreneurial intention, and entrepreneurship education (Table 13), with entrepreneurship education having the highest total link strength (655, Table 6). Moreover, personality traits (cluster 3), motivation (cluster 4), creativity (cluster 4), innovation (cluster 1), self-efficacy (clusters 3 and 4), and attitude (cluster 1) play an essential role in EI, according to the literature published in this period.

**Table 13.** Clusters related to documents published on EI (2014–2022).

| Cluster 1 | Cluster 2 | Cluster 3 | Cluster 4 | Cluster 5 |
|---|---|---|---|---|
| Attitude | Culture | Personality traits | Creativity | EB |
| Education | EA | Self-efficacy | EM | Higher education |
| Entrepreneur | EB | Social capital | ESE | |
| EE | EI | SEI | Motivation | |
| EO | EE | SE | | |
| ES | Gender | | | |
| Innovation | | | | |
| Intention | | | | |

Legend: EA (entrepreneurial attitude), EB (entrepreneurial behavior), EE (entrepreneurship education), EI (entrepreneurial intention), EM (entrepreneurial motivation), EO (entrepreneurial orientation), ES (entrepreneurial skills), ESE (entrepreneurial self-efficacy), SE (social entrepreneurship), SEI (social entrepreneurial intention).

The application of entrepreneurial soft skills through creativity, motivation, and entrepreneurial attitude and behavior is vital for achieving organizational leadership, complemented by long-term corporate sustainability. These entrepreneurial-related aspects of EI increase social capital and the entrepreneurial orientation of prospective entrepreneurs. As a result, the instinctive behavior of the entrepreneurs is moderated and depends significantly on leadership abilities and skills to motivate teamwork, the capacity to develop technological products or services, and age [33], in agile organizations defined by "the ability to adopt change in anticipation of disruptions using soft skills building on imagination, creativity, and ingenuity to innovate to move forward in rapidly changing environments" [34] (p. 23) to reduce corporate risk. As a result, when disruptions impact nations and firms, agile organizations can face challenges caused by these disruptions more quickly and efficiently. Additionally, business stability and KPIs (key performance indicators) are less threatened, and stakeholders are more prone to support and invest in the firm in the global VUCA environments.

As shown in Figure 2, the keywords "perceived behavioral control", "entrepreneurial behavior", and "motivation" were the most recently linked to EI. As presented in Table 13, the 3 year average multiplied by 2.19 during the COVID-19 pandemic compared to 2014–2016, where entrepreneurship-related research was primarily guided by psychological-

related traits (Table 14), with entrepreneurship education having the highest total link strength (655, Table 6), thus validating P1.

**Table 14.** Clusters related to HF as a keyword (SCOPUS database).

| Cluster 1 | Cluster 2 | Cluster 3 | Cluster 4 |
|---|---|---|---|
| Education | Entrepreneurship | Ethics | Sustainability |
| Leadership | Human flourishing | Morality | SD |
| Meaningful work | Humanistic management | Social responsibility | |
| Positive psychology | Wellbeing | | |

Legend: SD (sustainable development).

*5.2. HF and Ethics*

Regarding P2, as shown in Figures 4 and 5, the item "ethics" plays a crucial role and grounds HF development. Initially and related to HF, ethics dealt with morality and leadership before being later linked to virtue, technology, and sustainable development. Later, the keyword "ethics" was associated with wellbeing, character, meaningful work, and sustainability (Figure 5). HF is related to ethical-related topics with the goal of "reaching optimal levels of human functioning associated with goodness, generativity, growth, and resilience" [35] (p. 1). In this ameliorating process, HF also links to "soft skills, including resilience—agility interactions and the talent challenges organizations face meeting demands for soft skills that impact the wellbeing of people, performance, and productivity in the workplace" [36] (p. 10). Associated with mindfulness, happiness, and proactivity, HF can be a synonym for the accomplished life and the alignment of just actions related to personal goodness [37]. Consequently, HF comprises the sustainable pursuit of core personal projects in life [38–40].

Conceived as a human right, HF is fundamental to human dignity [41]. When an individual (or firm) flourishes, happiness increases in the person (or organization), generating positive externalities in their environment and favoring the work and family environments. As a result, positive externalities emerge, backing the economy and society. Everyone has the right to happiness and success, especially in competitive environments marked by resilience and constant change, where education has a crucial role.

Connected to HF, resilience is another key to integrating HF in the individual, as resilience "is the capacity to adapt to change using soft skills that facilitate to bounce back from difficult situations to reach a better state than previous to the disruption" [34] (p. 23). Resilient organizations have more capacity to deal with critical situations, as they have, in general, a leader who heads the organization toward business success. The primary relationship among ethics, leadership, and morality can be observed in Figures 4 and 5.

Mainly valued through academic training and professional experience, human capital is increasingly complemented by the generation of professional networks created by people (relational capital) and expedites moving in structured complex organizations (structural capital). As a result, the sum of human, relational, and structural types of capital creates the so-called "intellectual capital" to increase organizational value in the firm.

By combining the ethic of nature and people/community, socioecological wellbeing is an organizing principle that ensures HF [42]. Flow-friendly work environments help individuals flourish, thus increasing labor productivity and satisfaction at work [10]. As a result, the individual's desire to achieve greater wellbeing is born. Consequently, HF is an antecedent of future EI, as higher HF indicators positively impact EI and entrepreneurial self-efficacy. Especially in disruptive times caused by exogenous shocks, entrepreneurs and human teams with good personal relationships are more resilient and optimistic. They dispose of a propensity for positive events that may influence their self-motivation to achieve goodness [17].

HF, as the entrepreneurial enhancement feature built on resilience and agility, mental health, and wellbeing, delves into the effects caused by the individual and society to solve pandemic challenges. Every successful entrepreneur must be clear about their life purpose

from the beginning. This purpose must link to the organization and goes much further than conceiving the company as a means of survival. In developing countries, entrepreneurs who envision their companies only as their means of livelihood tend to have high entrepreneurial mortality because they have not imbued their successors with the necessary entrepreneurial spirit to grow the organization. Furthermore, freedom and autonomy intertwine with financial security, especially when the MSME can create cash flow, reducing the need for external financing. Excessive indebtedness and economic weakness due to the difficulty in accessing traditional banking caused by a lack of guarantees and informality are among the leading causes of bankruptcy in MSMEs, especially in developing countries. Given the above, and as shown in Table 14 and Figures 4 and 5, the entrepreneurial behavior related to ethics, morality, and social responsibility (cluster 3) determines HF, thus validating P2.

*5.3. Unicorns and Entrepreneurial Ecosystems*

Regarding P3, as shown in Figure 8, in 2018, the main keywords used in the entrepreneurship-related literature were "ecosystem" and "leadership". This narrow vision quickly broadened with the inclusion of "innovation" and "disruptive innovation". Concerning unicorns, firms are strongly related to the business model used, disruptive innovation, innovation diffusion, and commerce, as shown in cluster 1 (Table 15).

**Table 15.** Clusters related to the unicorn as a keyword (SCOPUS database).

| Cluster 1 | Cluster 2 | Cluster 3 |
|---|---|---|
| Business model | Ecosystem | Entrepreneurship |
| Commerce | Innovation | Startups |
| DI | Leadership | Venture capital |
| ID | Startup | |
| Unicorns | | |

Legend: DI (disruptive innovation), ID (innovation diffusion).

Disruptive innovation is crucial for unicorns to succeed based on technology and cheap financing, especially in R&D (research and development)-related industries. Unicorns tend to dispose of nontechnological and technological innovations, as the unicorns' birth and consolidation may be explained by the sequentially intertwined occurrence of cognitive biases to affect entrepreneurial decision making, from which establishment and legitimization eventually emerge [43]. These cognitive biases determine business strategies, future valuation of firms, and stakeholders' expectations. Innovation may be nontechnological, such as organizational and marketing, and technological, such as product and process innovation [44]. Industries linked to ICTs, financial services, energy, consumer goods, and automobile sectors are more prone to disruptive innovation [45]. Consequently, achieving ICT-based efficient, precise, and transparent decision making is vital to benefit shareholders. As a result, the combination of entrepreneurial ecosystems (cluster 2) inserted into an efficient business model (cluster 1) endowed with disruptive innovation (cluster 1) and leadership (cluster 2) represents key success factors for born globals to be unicorns, thus validating P3.

## 6. Conclusions

To analyze the main trends of global research into the values attributed to EI, HF, and unicorns from 2014 to 2022, a bibliometric analysis of 3528 documents obtained from the SCOPUS database was carried out, of which 2434 documents (1973 papers, 217 book chapters, 141 conference proceedings, 36 reviews, 30 books, and 37 other publications) were in the BMA and EEF subject areas. The main findings of this study were as follows: (1) among the top 12 countries, the number of documents published in the European Union on EI (600) almost doubled those published in the United States (354); the United States (113) led the number of papers published on HF, and the number of documents published by BRICS (Brazil, Russia, India, China, and South Africa) countries (22) on Unicorns almost

equaled the number of documents published in the US (25); (2) research on EI during the core years of the COVID-19 pandemic (2019–2022) revealed a growing interest linked to entrepreneurship education and psychological traits; (3) ethics-related entrepreneurial behavior has historically supported the current HF development; (4) entrepreneurial ecosystems, leadership, and innovation are critical success factors for born globals to be unicorns; (5) there is a geographic disparity (Spain, India, and the United States) in the most cited authors for EI (Liñán, F., University of Seville, Spain) (Table 7), HF (Sen, A.K., India) (Table 9), and unicorns (Audretsch, D.B., Indiana University, USA) (Table 11).

*Limitations and Future Research*

One of the limitations of this paper was the exclusive use of the SCOPUS database. This gap could be addressed in future research by crossing different databases, such as Web of Science, EBSCO, DOAJ, Dialnet, Latindex, and Redalyc. Furthermore, future research trends on entrepreneurship seem to be related to aspects related to psychological factors emanated from HF, the importance of unicorns in emerging countries and China, and the relationship with EI related to aspects linked to disruptive innovation, venture capital, meaningful work, business sustainability, leadership, well-being, motivation, entrepreneurial behavior, and entrepreneurial skills.

**Funding:** This research received no external funding.

**Institutional Review Board Statement:** Not applicable.

**Informed Consent Statement:** Not applicable.

**Data Availability Statement:** Data supporting the findings of this study are available from the author on request.

**Acknowledgments:** The author acknowledges the highly valuable comments and suggestions that contributed to improvements to a previous draft of the paper.

**Conflicts of Interest:** The author declares no conflict of interest.

## Appendix A

**Table A1.** Authors—Clusters related to EI as a keyword (SCOPUS database).

| Cluster 1 | Cluster 2 | Cluster 3 | Cluster 4 | Cluster 5 |
|---|---|---|---|---|
| Acs, Z.J. | Ajzen, I. | al-Laham, A. | Bandura, A. | Anderson, R.E. |
| Audretsch, D.B. | Bird, B. | Fayolle, A. | Crick, A. | Black, W.C. |
| Autio, E. | Brannback, M. | Fiet, J.O. | Greene, P.G. | Fornell, C. |
| Baron, R.A. | Casrud, A.L. | Franke, N. | Hills, G.E. | Guerrero, M. |
| Bosma, N. | Chen, Y.W. | Gailly, B. | Kickul, J. | Hair, J.F. |
| Brush, C.G. | Conner, M. | Kennedy, J. | Marlino, D. | Larcker, D.F. |
| Cardon, M.S. | Fink, M. | Matlay, H. | McGee, J.E. | Podsakoff, P.M. |
| Davidsson, P. | Fishbein, M. | Miao, C. | Mueller, S.L. | Ringle, C.M. |
| Douglas, E.J. | Hay, M. | Nabi, G. | Seibert, S.E. | Sarstedt, M. |
| Frese, M. | Kautonen, T. | Qian, S. | Sequeira, J.M. | Urbano, D. |
| Gartner, W.B. | Klofsten, M. | Souitaris, V. | Wilson, F. | |
| Hofstede, G. | Koenig, M. | Van Praag, M. | Zhao, H. | |
| Honig, B. | Kolvereid, I. | Westhead, P. | | |
| Kuratko, D.F. | Krueger, N.F. | Zerbinati, S. | | |
| Locke. E.A. | Liñán, F. | | | |
| Lumpkin, G.T. | Moriano, J.A. | | | |
| Minniti, M. | Reilly, M.D. | | | |
| Obschonka, M. | Schlaegel, C. | | | |
| Patzelt, H. | Shapero, A. | | | |
| Rauch, A. | Sokol, L. | | | |
| Shane, S. | Stephan, U. | | | |
| Shepherd, D.A. | Van Gelderen, M. | | | |

**Table A1.** *Cont.*

| Cluster 1 | Cluster 2 | Cluster 3 | Cluster 4 | Cluster 5 |
|---|---|---|---|---|
| Thurik, R. | | | | |
| Venkataraman, S. | | | | |
| Verheul, I. | | | | |
| Welter, F. | | | | |
| Wiklund, J. | | | | |
| Wong, P.K. | | | | |
| Wright, M. | | | | |
| Zahra, S.A. | | | | |

**Table A2.** Authors—Clusters related to HF as a keyword (SCOPUS database).

| Cluster 1 | Cluster 2 | Cluster 3 | Cluster 4 | Cluster 5 |
|---|---|---|---|---|
| Bakker, A.B. | Acemoglu, D. | Beadle, R. | Gaus, G. | Alkire, S. |
| Csikszentmihalyi, M. | Arendt, H. | Bolton, S.C. | Keohane, R.O. | Kahneman, D. |
| Deci, E.L. | Bauman, Z. | Friedman, M. | Mill, J.S. | Nussbaum, M. |
| Demerouti, E. | Bowles, S. | Hayek, F.A. | Miller, D. | Ruger, J.P. |
| Diener, E. | Bruni, L. | MacIntyre, A. | Nussbaum, M.C. | Sen, A.K. |
| Dutton, J.E. | Foucault, M. | Marx, K. | Pettit, P. | |
| Dweck, C.S. | Fraser, N. | Maslow, A.H. | Rawls, J. | |
| Fredrickson, B.L. | Harvey, D. | Mele, D. | Yeoman, R. | |
| Inglehart, R. | Hirschman, A.O. | Moore, G. | | |
| Kasser, T. | Lawson, T. | Pirson, M. | | |
| Luthans, F. | MacIntyre, A.C. | Sayer, A. | | |
| Lyubomirsky, S. | Polanyi, K. | Senge, P. | | |
| Moscardo, G. | Sen. A.K. | Sison, A.J.G. | | |
| Pearce, P.L. | Stiglitz, J.E. | Smith, A. | | |
| Peterson, C. | Streeck, W. | Solomon, R.C. | | |
| Ross, G.F. | Sugden, R. | Steffen, W. | | |
| Ryan, R.M. | Taylor, C. | Weber, M. | | |
| Ryff, C.D. | Van Lenge, P.A.M. | | | |
| Schaufeli, W.B. | | | | |
| Seligman, M.E.P. | | | | |
| Veenhoven, R. | | | | |

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
