# Peer review of "Network Data Maps on Entrepreneurial Intention, Unicorns, and Human Flourishing on the SCOPUS Database: A Visual Analysis Using VOSviewer"

_world, doi:10.3390/world3040045_

Round 1

Reviewer 1 Report

The topic of the paper is clear and current. The structure of the paper is well organised. The research method used in the paper is very simple. It should be described more detailed.

I recommend including the part concerning the conclusions after the discussion part.

Author Response

Please, see the file attached. Thanks in advance.

Reviewer 2 Report

This paper focuses on a very relevant theme. The Introduction provide an overview of the study. The Table 1. mentioning an Introduction is too early. The concepts are defined below in the 2. Literature review.The substantiation for the hypotheses do not correspond to the hypotheses formulated. I do not understand how you consider the Hypothesis 1 (H1): “During the core years of the COVID-19 pandemic (2019-2022), 111 researching entrepreneurial intention has been a growing interest.” as a research hypothesis. It's a fact!

The paper is a Bibliometric Analysis must present improvements in terms of Research Protocol, Analysis of Results and Discussion, presenting Clusters. The article should highlight the suggestions for future research of the analyzed articles and add other categories by clusters. It should highlight the theoretical and practical contributions, as well as the limitations of the investigation.

Author Response

Please, see the file attached. Thank you for your suggestions.

Reviewer 3 Report

General comment:

The paper provides a solid systematic literature review on entrepreneurial intention, human flourishing, and unicorns. Nevertheless, there are several formal aspects that can be improved in a new version of the manuscript, especially, in terms of bibliometric and co-citation analysis, as well as the inclusion of propositions and revision of the concluding remarks, and future research agenda.  

In order to improve the global quality of the manuscript, please consider the comments and suggestions provided below:

Specific comments:

1)     The motivation for producing the current systematic literature review needs to be positioned in terms of the ongoing debate on related literature devoted to entrepreneurial ecosystems.

2)     The aims of the paper need to be clarified.

3)     In the introductory item, please introduce a paragraph presenting the structure of the paper.

4)     Due to the systematic nature of the literature review, it is suggested to elaborate on propositions instead of hypotheses. In the discussion section, those propositions should be contrasted with previous literature.

5)     In the scope of the empirical analysis, it is suggested to provide a table with the research protocol (including both exclusion and inclusion criteria); a table with the most cited works and authors; as well as a co-citation analysis in order to assess the pathway of related publications. Examples of methodological designs that could be used in the revision of the manuscript can be found here:

DOI: 10.3390/su13052566;

DOI: 10.1080/14778238.2020.1751571

6)     It is recommended to include a new final section presenting the conclusions and a future research agenda.

Author Response

Please, see the file attached. Thanks a lot for your highly valued suggestions for improving the paper.

Round 2

Reviewer 2 Report

Accept in present form. Congratulations on the work done!

Reviewer 3 Report

Congratulations on the revised version of the manuscript.